# Resource selection of a montane endemic: Sex-specific differences in white-bellied voles (*Microtus longicaudus leucophaeus*)

**Neil R. Dutt**[1☺]*, **Amanda M. Veals**[2☺], **John L. Koprowski**[1☺¤]

**1** School of Natural Resources and the Environment, University of Arizona, Tucson, Arizona, United States of America, **2** Texas A&M University-Kingsville, Kingsville, Texas, United States of America

☺ These authors contributed equally to this work.
¤ Current address: Haub School of Environment and Natural Resources, University of Wyoming, Laramie, Wyoming, United States of America
* neildutt89@gmail.com

**Data Availability Statement:** All relevant data are within the manuscript and it's Supporting Information files.

**Funding:** NRD- Funding was provided by University of Arizona (http://www.arizona.edu/) and

## Abstract

Resources that an individual selects contrasted against what is available can provide valuable information regarding species-specific behavior and ecological relationships. Small mammals represent excellent study organisms to assess such relationships. Isolated populations that exist on the edge of a species' distribution often exhibit behavioral adaptations to the extremes experienced by a species and can provide meaningful insight into the resource requirements of the species. We deployed radio transmitters in a peripheral population of the long-tailed vole (*Microtus longicaudus)* during the mating season. We developed models of resource selection at multiple scales (within home range and patch). We found voles generally selected areas close to water and roads and consisting of high understory vegetation primarily composed of grasses. Resource selection varied between sexes suggesting different resource needs during the breeding season. The differential resource needs of voles might be a result of the energetic requirements for reproduction and are representative of a promiscuous or polygynous mating system.

## Introduction

Patterns of animal spatial distribution can have profound implications for conservation and management of species and their habitat [1]. Understanding patterns of resource selection can provide fundamental information about species ecology and their resource requirements that can inform current ecological knowledge and management strategies [1–4]. Differential habitat selection by individuals for the highest quality areas available can be observed in individual home ranges [5,6]. An individual's home range is often defined by multiple abiotic and biotic features of the environment and can shift based on the dynamic characteristics of these features and an individual's physiological requirements [7–9]. Which resources an individual selects within its home range can therefore provide valuable information about species-specific behavior and ecological relationships [9,10].

Resource selection functions are the most common way that resource use is evaluated from individual location data [2,11]. Resource selection functions are fit in a use–availability

T&E Incorporated's Grants for Conservation Research (www.tandeinc.com). The funders had no role in study design, data collection and analysis, decision to publish, or preparation of the manuscript.

**Competing interests:** We received funding from the commercial source, T&E Incorporated, for this project; this does not alter our adherence to PLOS ONE policies on sharing data and materials.

framework, whereby environmental covariates (e.g., elevation, distance to water) at the site the animal was located (the used locations) are contrasted with covariates at random locations taken from a realistically available area for selection (the available sample; [2,11]). Evaluating space use at multiple spatial scales provides detailed characterizations of species habitat profiles and informs management practices [2,12–14]. Resource selection can vary based on sex due to different strategies employed by either sex depending on mating system [15,16]. During mating seasons, marked differences exist in the movement and resource selection between sexes, particularly in promiscuous and polygynous species [17,18]; in these systems, individuals allocate limited resources to reproduction. In promiscuous and polygynous mating systems, males optimize their reproductive efforts by copulating with as many females as possible, whereas females maximize their reproductive efforts by obtaining and converting food into offspring [16]. Differential resource selection based on sex is driven by differing resource requirements determined by mating systems or strategies [15,19]. Identifying the resource needs at the individual level will better inform management decisions to preserve vulnerable populations.

Small mammals are an essential component of many ecosystems by providing vital ecosystem services [20,21]. Small mammals consume invertebrates, vegetation, and seeds, potentially aiding local plant communities by controlling damaging insect populations and influencing seed distribution [20,22]. Many predators specialize on small mammals in their diet [20,21]. Some fossorial and semi-fossorial small mammals, such as prairie dogs (*Cynomys* spp.) and kangaroo rats (*Dipodomys* spp.), are keystone species that fulfill a crucial role in an ecosystem's maintenance and health through bioturbation and habitat alterations [20,23,24]. Understanding the biology and ecology of small mammal populations is vital to inform conservation and management of ecosystems.

The long-tailed vole (*Microtus longicaudus*) is a small semi-fossorial rodent and habitat generalist [25]. The long-tailed vole boasts one of the largest latitudinal distribution ranges of all vole species in North America, stretching from Alaska to Arizona [25,26]. The importance of rear-edge populations, that is the populations that exist in the lower latitudes of widely distributed species, are often undervalued [27]. These edge populations can be a model that presents a focused view of how core populations may react and adapt to further species range contraction and expansion [28,29]. Like all microtines, long-tailed voles are herbivores with diets primarily consisting of leaves, stems, and roots of herbaceous plants [30]. The mating season for this widely distributed vole varies throughout the species range; populations at lower latitudes have an extended mating season from May–October with most reproductive activity in June–July [25]. *Microtus* presents a spectrum of mating systems from promiscuous (*M. pennsylvanicus*), polygynous (*M. xanthognathus*), and sometimes monogamous (*M. ochrogaster*); some species display one, two, or all three systems depending on the social environment and resource availability [17]. The mating system of long-tailed voles is not well understood, but likely exists within the spectrum displayed by congeners. The white-bellied long-tailed vole (*M. l. leucophaeus*; hereafter referred to as "white-bellied vole"), a sub-species of the long-tailed vole, is endemic to the Pinaleño Mountains in southeastern Arizona [31]. This is the southernmost population of *M. longicaudus* and lacks informative research. Unlike many other populations of long-tailed voles, *M. l. leucophaeus* does not co-occur with other vole species, making this population particularly unique [31]. Whereas most other species in the genus inhabit areas dominated by grassy cover, the long-tailed vole is found in a variety of habitats ranging from typical grassy areas to sparsely vegetated or woody shrub areas [25,32]. Long-tailed voles are often less aggressive than other vole species and, as a result, are relegated to less favorable habitat [25,32]. The Pinaleño Mountains population presents a unique opportunity to study resource selection in the absence of other vole species that may relegate long-tailed voles to

sub-optimal habitat [31]. Furthermore, white-bellied voles are a Species of Greatest Conservation Concern in the State of Arizona [33].

In this study, we examined how white-bellied voles use the landscape available in the absence of congeneric competition at multiple spatial scales. We modeled resource selection within the home range (3rd order; [34]) and for specific fine scale resources (4th order; [34]). Our objectives were to model fine scale patterns of resource selection for a peripheral population of long-tailed voles and examine differences between the sexes. We predicted that white-bellied voles will select areas with herbaceous grassy cover over areas dominated by woody vegetation. We predicted that males and females would show different patterns of resource selection across spatial scales due to differing energetic needs (i.e. proximity to water or forage) and reproductive strategies (i.e. nest sites).

## Materials and methods

### Study area

The Pinaleño Mountains (32.7016˚N, -109.8718˚E) are a portion of the northern extent of the Madrean Sky Island Complex of the southwestern United States [35]. At 3,269 m the Pinaleño Mountains have the highest peak in the complex and encompass an area of approximately 780 km². The Pinaleño Mountains have diverse vegetation communities as a result of the 2,367 m elevational gradient [31]. Forest landscapes in the Pinaleño Mountains have been fragmented due to roads, insect outbreaks, and large-scale fires. These high levels of disturbance resulted in habitat that is extremely patchy and poorly connected [36]. Our study area was located in the upper elevations of the range, consisting of high mountain meadows and mixed conifer forests (2,870–3,050 m) of Douglas-fir (*Pseudotsuga menziesii*), ponderosa pine (*Pinus ponderosa*), southwestern white pine (*Pinus strobiformis*), Engelmann spruce, and corkbark fir (*Abies lasiocarpa* var. *arizonica*) [37]. We conducted our study in 9.1 ha of meadow and mixed conifer forest.

### Sampling design

We trapped small mammals during the summers of 2018 (May–August) and 2019 (May–August) over multiple sessions. We trapped areas that were > 100 m apart and between 50 and 500 m from roads to avoid any negative road impacts, such as avoidance or mortality [38]. We used single door folding Sherman traps (7.62 x 8.89 x 22.86 cm; H.B. Sherman Traps, Inc. Tallahassee FL) baited with a mixture of peanut butter and oats.

To determine areas where we could reliably trap white-bellied voles we conducted preliminary trapping in 2018 based on historical locations and information from previous studies [31,39,40]. For each trapping session, we placed two transects that were open both day and night due to the lack of clarity on activity periods of long-tailed voles (diurnal [31]; nocturnal [25]). Our transects were 240 m long and located parallel to Soldier Creek and an un-named creek near Coronado National Forest access road 4567. Along each transect, we placed a pair of traps every 10 m, one on either side of the watercourse, for a total of 50 traps. We placed each pair of traps in dry locations 1–5 m from the watercourse depending on saturation of the soil, as some areas were diffused and bog-like. In 2018, we opened traps at dusk and checked them approximately 30 min after dawn. After the morning check, we left the traps open, and checked and closed them at 1000 h to avoid heat related mortalities. We reopened traps in the afternoon and checked the traps again at dusk. We discontinued nocturnal trapping after 2018 as diurnal trapping proved to be more successful. In 2019, we placed transects where voles were caught the previous year. Additionally, we opportunistically trapped in areas where we observed voles in 2019.

## Animal handling

We processed and released animals immediately at the location of capture. We characterized age class (juvenile or adult), sex, and reproductive condition by visual inspection of testes (scrotal, abdominal) and teats (lactation) or vaginal condition (perforate, nonperforate). We recorded mass in g, and following standard marking methods, we used ear tags (1005–1, National Band and Tag Company, Newport, KY; Sikes et al. 2016 41) to mark individuals. We affixed very high frequency radio collars (SOM-2070, Wildlife Materials, Murphysboro, IL) to 31 adult white-bellied voles (body mass $\geq$ 30 g, mean $\pm$ $SD$ = 56.2 $\pm$ 7.2 g) in 2019. Radio collars were < 10% of each individual's body mass (mean = 3.09 $\pm$ 0.09 g) to minimize effects on daily activity and behavior [41,42].

## Radio telemetry

We located animals five of the seven days per week, with an animal receiving several points throughout the day in May–September 2019. We obtained locations >1 h apart, distributed across daylight hours to ensure temporal independence of locations [43]. Animals were located by homing on individuals until one of the following occurred: 25 telemetry locations were achieved [44], the collar fell off, the vole was predated, or the collar's signal could not be located. We used a handheld global positioning system device and used the point averaging function for $\geq$ 5 minutes to record the spatial location of all animal points.

All field work was conducted in accordance with the American Society of Mammalogists guidelines [41] and approved by the University of Arizona's Institutional Animal Care and Use Committee (IACUC protocol #16–169) under permit from Arizona Game and Fish Department (Permit # SP651773).

## Vegetation sampling

Upon locating a vole via telemetry, we marked the location with a pin-flag. To minimize the effect of our presence on vole movement we waited one hour to collect vegetation data at the location, given the vole had moved away from that location. We collected vegetation data at two points: the vole's known location and at a paired, randomly generated location [14,45]. Random locations were placed 9.8 m away from the vole's known location, which we based on the average hourly movement rate of other vole species [42,46–48]. We used a random number generator to represent eight intermediate and cardinal directions moving clockwise. At all known and random locations, we recorded understory cover, canopy cover, vegetation composition (categories: bare ground, coarse woody debris, grass, forb, fern, log, sedge, stump, rock, rush, shrub, tree, water). We used a 2.5 cm x 100 cm cover pole marked in 2 cm increments to measure understory cover at the center of each location [49]. We recorded the height of any obscuring vegetation from the four cardinal directions at a distance of 4 m and a height of 1 m, with any vegetation taller than 80 cm classified as 100% understory cover. We calculated percent understory cover by taking the average from all four measurements at each location, divided by the total height of the cover pole [49]. To calculate canopy cover we followed the standard equation for a convex spherical densiometer and applied the correction factor for the 17-dot variation for each point [50]. We used a 1 m$^2$ quadrat centered at each location to characterize vegetation composition and percent cover through visual inspection [51]. For any woody species within the 1 m$^2$ quadrat, we used diameter at breast height (DBH) to categorize live and dead woody species as either shrubs (woody plants < 10 cm DBH) or trees ($\geq$ 10 cm DBH [52]).

## Data analysis

We calculated 95% minimum convex polygons using the "adehabitatHR" package in R [53,54], and visualized estimates with ArcGIS Pro v2.4.1 [55] for all individuals that had at least five telemetry locations, the minimum number of locations needed by the "adehabitatHR" package to create an individual minimum convex polygon home range. The number of locations per individual ranged 5–25 (mean = 18). We pooled all animals into three groups: all voles, males, and females; pooling data across individuals while still accounting for individuals variation is ideal for low sample sizes [56]. In total, we used 493 known locations, 325 from 19 females and 168 from nine males. There was heterogeneity in the number of locations per individual, so to define availability at the within home range scale (3rd order), we generated random points at a 2:1 ratio, within each individual's home range, to ensure availability was unique to each individual [56]. For all animal locations and random points, we extracted normalized difference vegetation index (NDVI) values, distance to roads, and distance to water in ArcGIS Pro. The remotely sensed imagery we used was taken in late summer 2017 and is the most recent and highest resolution (30 cm) available [57]; we overlaid this imagery with road and waterway layers and hand digitized where needed.

We used the paired random locations (9.8 m from known locations) taken in the field to define availability for patch scale (4th order) selection. We modeled selection based on habitat features at known vole locations compared to the corresponding random location, where we measured habitat characteristics simultaneously to remove effects of differing availability by weather and time of day [58].

We quantified resource selection of voles at multiple scales to assess habitat selection and identify key environmental characteristics. We standardized all covariates prior to running models. We fit generalized linear mixed-effects logistic regression models with individual as a random effect and a binomial use vs. availability design, with the lme4 package in R for within home range scale selection. To reduce bias based on unequal known locations, we used a random intercept term assigned to each individual [56,59]. At the patch scale, to compare each vole location with its random location, we used conditional mixed-effects logistic regression models using the mclogit function in R with a binomial error structure and logit link function [2,11,56]. We tested sets of *a priori* models at both scales based on previous research of habitat selection of long-tailed voles [25,31]. We tested models at both scales for three groups: all voles, males, and females (within home range: 2 models; patch scale: 5 models). We evaluated model support using Akaike's Information Criterion adjusted for small sample sizes (AICc), and all models ≤ 2 AICc units of the top model were considered to be competing models [60,61].

## Results

### Captures

We logged 4,715 trap nights (3,300 in 2018 and 1,415 in 2019) and captured 194 individual voles in total: 45 unique voles in 2018 (17 males, 20 females, 8 juveniles) and 149 unique voles in 2019 (31 males, 55 females, 63 juveniles). In 2019, we collared and tracked 31 adult voles (≥30 g; 12 males, 19 females). To collect animal locations via radio telemetry, we logged approximately 558 person hours. More than half (58%; 7 males, 11 females) of these individuals were either lost from the study due to predation (4 males, 1 female), unknown cause but confirmed mortality (2 females), collar removal (1 male, 8 females), or no signal/collar malfunction (2 males); on average, loss of a vole (i.e. no longer able to have data collected) would occur 29.83 ± 20.69 (SD) days after receipt of collar.

## Within home range selection

At the within home range scale, our top model for each group indicated negative selection (i.e. avoidance) by white-bellied voles for high NDVI values and areas farther from roads (Fig 1). For the all voles (sexes combined) group, our top model included all covariates: NDVI, distance to water, and distance to roads (Table 1). The top model for females included NDVI, distance to roads, and distance to water and the top model for males included NDVI and distance to roads (Table 1). For all voles and females, top models indicated selection for areas farther from water. The top model for males excluded distance to water and indicate avoidance of high NDVI values and areas farther from roads. However, our three groups each had two competing models that were within 2 ΔAIC units (Table 1).

## Patch level selection

For patch scale, the all voles, female, and male groups had the same top model which included: understory cover, canopy cover, bare ground, grass, forb, logs, coarse woody debris, distance to water, and distance to roads (S1 Table). The beta coefficients, for the all voles top model, indicated positive selection for high understory cover, canopy cover, grass, forb, logs, and coarse woody debris and avoidance of bare ground, distance to water, and distance to roads. White-bellied voles selected for areas with high understory cover and coarse woody debris (Fig 2). The white-bellied voles selected areas close to roads, however, we never documented voles

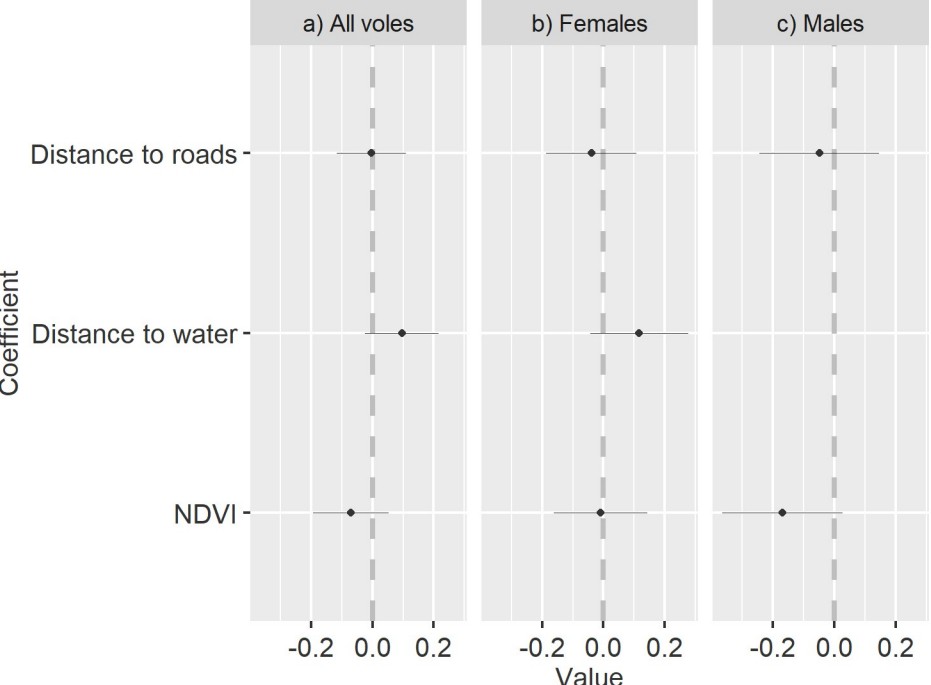

**Fig 1. Home range scale beta coefficients.** Habitat selection patterns of white-bellied voles (*M. l. leucophaeus*) from the Pinaleño Mountains in southeastern Arizona, USA, summer 2019 represented by beta coefficients of variables explaining variation in habitat selection patterns of our top 3rd order generalized linear mixed-effects logistic regression model for a) all voles, b) Female, and c) Male voles. The x-axis depicts standardized regression coefficients, which provide an index of the strength of the linear relationship for explaining habitat selection patterns. The y-axis contains all the covariates included. The dotted line at zero represents the division between selection (right of line) and avoidance (left side of line). The coefficient estimates are represented as dots and their 95% confidence intervals as whiskers.

**Table 1. Within home range scale (3<sup>rd</sup> order) *a priori* generalized linear mixed-effects logistic regression models.**

| Group | Covariates | AIC | ΔAIC |
|---|---|---|---|
| **All voles** | NDVI+Distance to roads+Distance to water | 1840.3 | 0.0 |
| | NDVI+Distance to roads | 1840.8 | 0.5 |
| **Females** | NDVI+Distance to roads+Distance to water | 1239.5 | 0.0 |
| | NDVI+Distance to roads | 1239.7 | 0.2 |
| **Males** | NDVI+Distance to roads | 606.0 | 0.0 |
| | NDVI+Distance to roads+Distance to water | 607.1 | 1.1 |

NDVI, normalized difference vegetation index; AIC, Akaike information criterion.

ΔAIC is the difference in AIC values between each model and the lowest AIC model.

crossing roads and only 2% of recorded locations were ≤ 10 m of roads. Of the three individuals that we documented ≤ 10 m from roads, none moved closer than 4 m to a road. Our model for female voles had positive beta coefficients for understory cover, canopy cover, logs, and coarse woody debris but negative beta coefficients for bare ground, grass, forb, distance to roads, and distance to water. Females strongly selected for areas with high understory cover and avoided areas of bare ground (Fig 2). We had a competing top model for females that included Grassy Cover as a covariate (ΔAIC = 0.6). Our male model indicates positive selection for all covariates except bare ground. Males strongly selected for areas with high log and coarse woody debris cover (Fig 2).

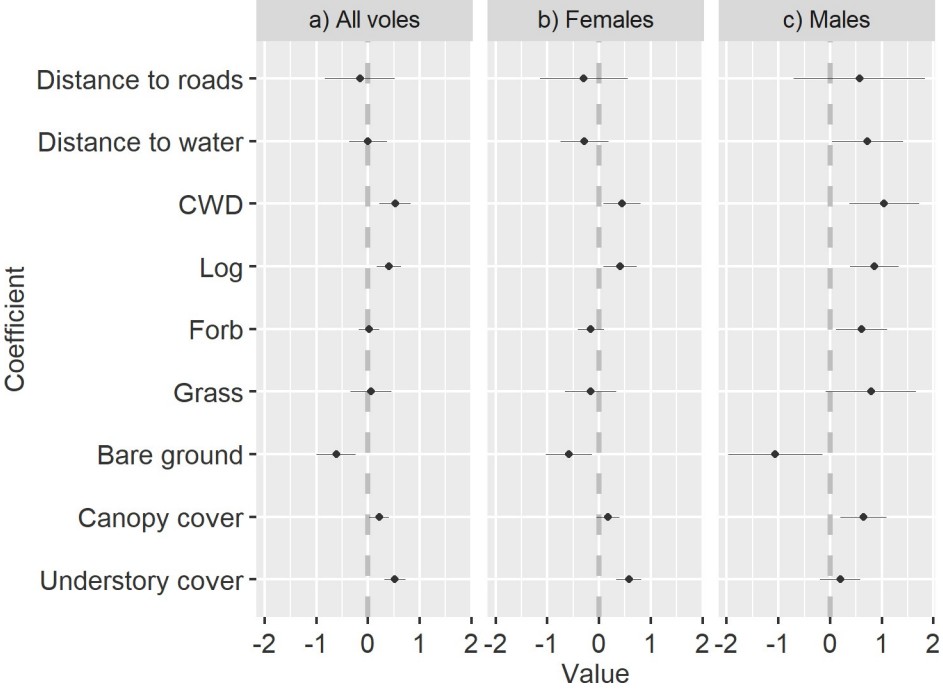

**Fig 2. Patch scale beta coefficients.** Habitat selection patterns of white-bellied voles (*M. l. leucophaeus*) from the Pinaleño Mountains in southeastern Arizona, USA, summer 2019 represented by beta coefficients of variables explaining variation in habitat selection patterns of our top 4<sup>th</sup> order conditional mixed-effects logistic regression model for a) All voles, b) Female, and c) Male voles. The x-axis depicts standardized regression coefficients, which provide an index of the strength of the linear relationship for explaining habitat selection patterns. The y-axis contains all the covariates included; coarse woody debris is shortened to CWD. The dotted line at zero represents the division between selection (right of line) and avoidance (left side of line). The coefficient estimates are represented as dots and their 95% confidence intervals as whiskers.

## Discussion

### Within home range selection

Resource selection is similar between the sexes of white-bellied voles. At the within home range scale, both sexes avoided areas with higher NDVI values corresponding to more heavily wooded areas. This indicates selection for more open areas with low tree cover. Our study area is a mosaic of mostly closed canopy forest and open grassy meadows as well as areas of patchy tree canopy cover with an understory consisting of bare ground, forbs, ferns, and grasses. Our data contrast with some previous studies of long-tailed vole habitat selection for wooded or shrubby areas and may be attributed to the lack of congeneric competition [25,32,62].

Both sexes selected areas relatively close to roads, however, we did not document any individuals that crossed a road. Several other small mammals avoid road surfaces due to the perceived threat of predation from the lack of cover [38]. Other highly mobile mammals such as squirrels exhibit clear avoidance of roads and rarely cross them [63]. Road type, size, and traffic volume are all factors that impact animal behavior and perceived risk [64]. The roads in our study area are hard compact, two-lane roads and are likely a barrier to vole movement. Selection for areas close to roads may be an artifact of roads occurring near flat open meadows where they are easier to construct [65] and voles selecting for roadside habitats due to an abundance of vegetation [66].

We found differences between the sexes in selection, at the within home range scale, of distance to water; the top model for females included distance to water and the top model for males did not. In mammals, food and water are common limitations for female fitness due to the increased energetic costs of pregnancy and lactation, whereas male fitness is limited by access to mates [15,67,68]. Vegetation community composition and predator avoidance are likely more important to males and resource selection may not be limited by water availability. Dependence on surface water by long-tailed voles has not been verified; some studies found surface water to be non-essential to long-tailed vole diets [25,69] whereas Findley et al. [70] found surface water is required for daily survival. It is possible that dependence of long-tailed voles on water is dictated by the type and quality of forage an individual consumes.

### Patch level selection

Despite selecting for similar areas overall, males and females displayed variability in selection patterns at the patch scale. Our All voles model indicates white-bellied voles avoid areas of bare ground and areas far from roads and water. The All voles model displays varying degrees of positive selection for all other covariates; logs, coarse woody debris, and understory cover were most heavily selected. White-bellied voles select areas that have high amounts of cover with the cover types downed logs and coarse woody debris being the most highly selected for. This aligns with previous research that has found a positive correlation between number of logs and high vole densities [32]. Where our findings differ from previous research is the association with areas of sparse herbaceous growth. Long-tailed voles are known to use areas consisting of primarily woody vegetation in the presence of other vole species [32,62]. White-bellied voles strongly avoided bare ground and selected areas with high herbaceous understory cover in our study, which is consistent with findings from vole removal and exclusion experiments [62]. Our empirical evidence further strengthens the conclusion that long-tailed voles inhabit and flourish outside of the 'typical' habitat dominated by woody vegetation [62]. In the absence of competing vole species, white-bellied voles can select highly productive herbaceous areas for forage, without impediment, while still staying close to areas with logs and coarse woody debris for nest sites and predator evasion.

In promiscuous and polygynous systems, females tend to spend more time in creation and maintenance of burrows whereas males travel from female to female and attempt to mate and defend their home range from other males [17]. When we evaluated resource selection by sex, the different priorities during the mating season (May–October: [25]) become apparent. Females avoid bare ground, areas far from roads, and areas far from water as well as areas with high percentage of grass and forbs. Females select areas with high understory cover consisting primarily of grass for forage and cover but strongly select areas suitable for nests with micro-habitats consisting of logs and coarse woody debris near water. Females spend most of their time in suitable nest sites and only make short, infrequent trips into highly productive herbaceous areas to forage and gather nesting materials. Female voles require supplemental surface water sources outside of water obtained through herbaceous forage during times of high energetic requirements such as lactation [67].

In contrast, males avoid bare ground but selected areas farther from roads and water. Males selected a greater diversity of habitat characteristics, which suggests more movement within their home range. That males expend much of their time and energy to secure areas of abundant resources and therefore mating opportunities while excluding other males is indicative of resource defense polygyny [15]. *M. ochrogaster* and *Myodes gapperi* under non-drought conditions do not require supplemental water outside of water obtained from forage [67,71]. During the mating season, the highly mobile male white-bellied voles may avoid areas close to water because they consume forage with high moisture content that circumvents dependence on surface water as displayed by congeners [12,46]. An alternative explanation to males selecting areas far from water during the mating season is that males reduce water consumption as a tradeoff for more reproductive interactions similar to many promiscuous and polygynous ungulates during the breeding season [72,73]. Females are more sedentary and select areas near surface water to supplement their limited forage opportunities and increased energetic needs. Our findings of sex-based differential resource selection are consistent with previous research on other mammalian promiscuous and polygynous species [16,74].

## Conclusions

White-bellied voles' selection within home range for herbaceous vegetation as opposed to the woody habitat described for this species can help to inform future species management decisions. However, further research that incorporates additional variables at the within home range scale in conjunction with our results may be necessary. Given the potential importance of white-bellied voles in the Pinaleño Mountains, the ecosystem services they provide [23,25] and their imperiled conservation status in Arizona, it is crucial to understand the resource requirements and assess response to a changing climate to maintain this endemic population. By understanding the different patterns of resource selection for this subspecies in contrast to other populations will lead to better informed and more successful management decisions and illuminate key drivers of the species' biology. Because environments inherently change through time, it is important to understand the foundation of what individuals require and how they behaviorally respond to such change. Resource selection functions allow us to quantify resource selection patterns by wildlife [59]. Identifying these patterns and how they change through time can provide crucial insights into underlying resource needs of wildlife populations to inform management and conservation.

## Supporting information

**S1 Table. Patch scale *a priori* 4th order conditional mixed-effects logistic regression models.**
(DOCX)

## Acknowledgments

Thank you to R. Steidl, M. Merrick, M. Mazzamuto, and V. Greer for additional assistance with this project. Thank you to research technicians and volunteers that assisted with this project: M. Gilboy, B. Dobroslavic, K. Thacker, A. Dixson, B. Blais, C. Brocka, M. Morandini, B. Mayer, C. Shaw, S. Slovikosky, and D. Ziegler.

## Author Contributions

**Conceptualization:** Neil R. Dutt, Amanda M. Veals, John L. Koprowski.

**Data curation:** Neil R. Dutt.

**Formal analysis:** Neil R. Dutt.

**Funding acquisition:** Neil R. Dutt, John L. Koprowski.

**Methodology:** Neil R. Dutt, Amanda M. Veals, John L. Koprowski.

**Project administration:** John L. Koprowski.

**Supervision:** John L. Koprowski.

**Visualization:** Neil R. Dutt.

**Writing – original draft:** Neil R. Dutt.

**Writing – review & editing:** Amanda M. Veals, John L. Koprowski.

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
