## [Decision Letter · Decision Letter 0]

25 Sep 2020

PONE-D-20-27373

Resource selection of a montane endemic: sex-specific differences in white-bellied voles (Microtus longicaudus leucophaeus)

PLOS ONE

Dear Dr. Dutt,

Thank you for submitting your manuscript to PLOS ONE. After careful consideration, we feel that it has merit but does not fully meet PLOS ONE’s publication criteria as it currently stands. Therefore, we invite you to submit a revised version of the manuscript that addresses the points raised during the review process.

We look forward to receiving your revised manuscript.

Kind regards,

Bi-Song Yue, Ph.D

Academic Editor

PLOS ONE

2.Thank you for stating the following in the Financial Disclosure section:

[NRD- Funding was provided by University of Arizona (http://www.arizona.edu/) and T&E Incorporated’s Grants for Conservation Research (www.tandeinc.com). The funders had no role in study design, data collection and analysis, decision to publish, or preparation of the manuscript.]. 

We note that you received funding from a commercial source: [T&E Incorporated]

Reviewers' comments:

Reviewer's Responses to Questions

**Comments to the Author**

1. Is the manuscript technically sound, and do the data support the conclusions?

Reviewer #1: No

Reviewer #2: Yes

2. Has the statistical analysis been performed appropriately and rigorously? 

Reviewer #1: I Don't Know

Reviewer #2: Yes

3. Have the authors made all data underlying the findings in their manuscript fully available?

Reviewer #1: No

Reviewer #2: Yes

4. Is the manuscript presented in an intelligible fashion and written in standard English?

Reviewer #1: Yes

Reviewer #2: Yes

5. Review Comments to the Author

Reviewer #1: Review PONE-D-20-27373

This manuscript presents results of an interesting study on habitat selection by an endemic subspecies of long-tailed vole. The authors used radio telemetry to determine used locations, which were compared to available locations. At the 3rd order scale they compared GIS variable, while at the 4th order scale they compared field collected vegetation data. Research on habitat selection of poorly understood species such as this are much needed. However, I found a number of problems with the presentation of the information that made it very difficult to understand the methods and results. For instance, the authors ostensibly investigated habitat selection at the “within home range” and “patch” scale (3rd and 4th order). However, they consistently referred to the 3rd order as home range selection, which is actually 2nd order selection per Johnson (1980). The methods do not say how the available locations were selected for this scale. The microhabitat data should probably also be considered 3rd order selection (usage of various habitat components within the home range. Fourth order would normally be for specific items such as den sites or specific food items. So, I think both scales analyzed are actually 3rd order.

Another problem is that they concluded that their results reflected fundamentally different habitat selection for long-tailed voles in the Pinaleno Mountains, in comparison with other populations where long-tailed voles live in presence of other competitively dominant vole species. There is no evidence for this conclusion. Importantly, they conflate their results at the 3rd or 4th order of selection with generalized statements about habitat at the 1st or 2nd order. You cannot extrapolate from one scale to another. Second, the habitat conditions described by their results are consistent with other places that long-tailed voles are known to occur, including places with other vole species. To make such a comparison they would need to have similar habitat selection results for another population of voles where other species also exist.

The discussion was very difficult to follow as it seemed to jump all over the place often not identifying what scale was being discussed. I had a very difficult time following the arguments in that section due to the writing problems. I suggest a more streamlined discussion that more concisely describes 3rd order habitat selection in this population voles (without conclusions about completion) and describes the differences in habitat selection between the sexes. Other points don’t seem very well justified to me.

38-42 There is a bit of redundancy in these sentences.

76 If you are talking about the geographic range, then you are discussing a feature of the species, not the individual. A species should be referred to with the Latin name or “the long-tailed vole”, which denotes the species. If you are talking about multiple individuals that actually exist out in nature then you can refer to long-tailed voles. It is not correct to use long-tailed voles to refer to the species.

82 Missing a period. Also, is there a reason to say monocots and dicots? What other herbaceous plants are there?

92 Populations in New Mexico are just as far south.

96 In the Southwest they seem to be found in relatively moist locations and are mainly riparian (e.g., Findley et al 1975), although they also occur in openings within mixed coniferous forest that allows some herbaceous cover (e.g., Lehmkuhl et al 2008 Northwest Science 82, Wampler et al. 2008 SWNAT 53, Sullivan and Sullivan 2018 Crop Protection 112).

97 Something wrong with the wording

100 missing hyphen in white-bellied.

107-109 But you don’t have a comparison with a situation where they occur with a congeneric competitor. You already made it clear that this population lacks another species of vole. I think you should delete “in the absence of congeneric competition”. I also don’t think your study design allows you to address this point. The most you can say is that your results reflect habitat selection at the third order in your study area. You don’t know how that would differ in presence of another species.

108-109 Can you state what you predict the difference will be?

114 According to the USGS place names database, Mount Graham is a specific summit of the Pinaleno Mountains. I suggest deleting Mt. Graham here unless one of your study sites is on Mount Graham, or work the name into the study area description another way.

129 What is meant by “negative road impacts”?

139 What do you mean by “depending on the saturation of the soil?

150 how did you get an accurate total length measurement on a live animal? Why were these measurements taken? What about mass?

157 A location once per day per animal?

159 The “collar was slipped” is wildlife slang; please rephrase. Can you rephrase to avoid word predated? How did you know they had been killed by a predator? Did you find them dead and partially consumed? The “Signal was lost” is slang.

159-160 It is still not clear to me if you recorded more than one location per animal each day.

157-161 How did you determine the location of the animal? Triangulation? Homing? Did you evaluate telemetry system error?

167 I don’t know how you located the vole. Did you see it? IF not, now can you be sure its location?

174 and 179 Is canopy cover the same as overstory cover?

174 I do not understand how the vegetation sampling was collected. Were all these variables collected at the location or on transects or a plot?

178 At what distance did you consider obscuring vegetation? It is not clear how you measured this.

180 Here you mention averaging 4 measurements but you did not explain how you took them.

181 Did you take the densitometer reading in the 4 cardinal directions?

183-184 centered at the location? How did you measure composition and percent cover?

184 What is the plot? Do you mean the 1 m square plot?

190 Why did you include animals with so few locations?

194 I think you mean within home range scale, at least that is what I think you said you were going to evaluate in the introduction.

194 where were the random points drawn from?

196 What date did NDVI come from and why?

197 What were the sorts of water sources in the remote sensed imagery. Often voles are associated with moist to wet spots that would not show up on imagery or maps. You trapped along creeks. Were they perennial?

205-206 I did not see mention of quadrates in the methods. Are these cover classes for percent cover?

211-220 To me it seems that both of your scales are really third order, selection of sites within the home range. Your patch scale seems like description of microhabitat. Your other scale seems like within home range (not home range), although I cannot determine where you drawing the available points from. If within home range, I think your available points should be drawn from the MCP of each individual.

214 within home range scale?

218-219 Where are the models? How many? How rationalized? The sentence implies the same models for each scale. Do you not expect different selection at different scales?

224-225 I assume the male and female numbers are adults. How did you assign age class? Did you collar juveniles? Did you record sex on juveniles?

228-229 clarify what you mean by unknown but confirmed mortality. I think you mean you know it was dead but don’t know why. Collar slipping is slang. I think you mean that the collar fell off the animal. What does missing mean? Do you mean that you were unable to find a signal?

230 What do you mean by loss of vole?

222 You used individual as random effect but I don’t know the sample size of male and females included in the analyses.

235 by group do you mean gender?

237-238 Avoid using “shows” and “showing”

249 I don’t think you mentioned standardizing these variables in the methods, although perhaps I missed it.

256 Ideally I’d like to see the PCA results perhaps in supplemental material.

255 So did you not use the PCA? I am confused by what is being presented in this section.

258 This sentence does not read correctly—too many “models”.

260-263 Why not say that the Beta coefficients indicated positive selection for … and avoidance for ….

Supplement Table 1. In looking at this table I am confused about your choice of variables and a priori hypotheses. Is the global model the top model? Are these all the models? I don’t think I understand why the PCI is used versus the other variables in the models? For instance, was log and grass also cover classes? I also see no rationale for these models.

287 what do you mean by they showed varying degrees of selection?

290-291 Long-tailed voles are commonly captured in wet meadows and other herbaceous systems even when other voles are present. I’m not sure you can make this conclusion. Your analysis was at the home range (or perhaps within home range) and patch scales, while the habitat data you are contrasting with pertains to landscape or macro scale. I do not think it is surprising that long-tailed voles select the herbaceous environments in their surroundings. The same is true elsewhere in their range. In reflecting on this and looking back at your study area and trapping transect locations I really cannot get a feel for the environment in which the study was done. Was a mosaic of forest and open areas?

297-298 There are a number of studies, mainly in the Midwest, that discuss the importance of roadside environments for voles and other species that select dense herbaceous vegetation. The idea is that these areas collect more moisture resulting in more lush vegetation.

286-308 long paragraph with lots of different topics

312-313 Selection for downed logs and coarse woody debris paints a different set of conditions than grassy areas.

315 long-tailed voles occur in relatively open areas. When they occur in forested ecosystems, their local distribution is in openings in the forest where there is higher herbaceous ground cover. Later, you also say that they select high herbaceous cover. You can only get high herbaceous cover if the sun is able to hit the ground, which does not happen in dense forests.

319-320 I don’t think you can say your results support the idea that they experience competitive exclusion from high quality vole habitat. You did not test that.

321-322. What is open habitat dominated by woody vegetation? I don’t understand what you mean by that.

323-324 This is a perfect description of the habitat of the long-tailed vole, but it holds true for almost any other mountain range where other species of voles also occur. I don’t think your description of habitat has to do with presence of the other species of vole. Otherwise, why would two species of vole so frequently co-occur together to the point that you catch them both on the same trap line?

You surveyed for voles along creeks in a generally forested area in the mountains. This is an ideal situation for long-tailed voles. It provides moist areas with herbaceous vegetation in openings and meadows within the forest. You can find long-tailed voles in any mountain range in this same conditions, regardless if there are other species.

338-339 It could also mean they select the location of home ranges based on different characteristics.

349 And higher water requirements during lactation

352-353 Again, I think you are confusing scales as I don’t see your results different from other locations where the species occurs.

354 There is a danger of extrapolating third and fourth order selection to higher order selection such as geographic distribution. You don’t know what first and second order selection are.

358 Are they imperiled? SGCN directs funding as is not

Reviewer #2: This is a very interesting paper that examines space use in a less-studied species of vole, The All Voles model showed that voles avoided areas of bare ground and and areas far from roads and water, Females followed the same pattern with the addition of areas with high understory cover. Females tend to remain in their terriroty near thier nest sites. Males avoided bare ground but chose areas furthre fron road s and water. The authros try to fir their data to the promiscusous/polygynous mating systems, I found this fit most tenuos as the authros really have no direct data of interatiions between and within the sexes and their statments about mating systmes comes mainly from a revew chapter by Jerry Wolff (1985).

6. PLOS authors have the option to publish the peer review history of their article (what does this mean?). If published, this will include your full peer review and any attached files.

Reviewer #1: No

Reviewer #2: No

---

## [Author Response · Author response to Decision Letter 0]

20 Oct 2020

We appreciate the feed back from the Academic Editor and the reviewers on our manuscript. We have taken their comments into consideration and and responded to them in the document "Response to reviewers". We will also provide our responses here (below) to ensure that we are following procedure. Thank you for inviting us to resubmit this manuscript for potential publication.

2.Thank you for stating the following in the Financial Disclosure section:

[NRD- Funding was provided by University of Arizona (http://www.arizona.edu/) and T&E Incorporated’s Grants for Conservation Research (www.tandeinc.com). The funders had no role in study design, data collection and analysis, decision to publish, or preparation of the manuscript.]. 

We note that you received funding from a commercial source: [T&E Incorporated]

Reviewers' comments:

Reviewer's Responses to Questions

Comments to the Author

1. Is the manuscript technically sound, and do the data support the conclusions?

Reviewer #1: No

Reviewer #2: Yes

2. Has the statistical analysis been performed appropriately and rigorously?

Reviewer #1: I Don't Know

Reviewer #2: Yes

3. Have the authors made all data underlying the findings in their manuscript fully available?

Reviewer #1: No

Reviewer #2: Yes

4. Is the manuscript presented in an intelligible fashion and written in standard English?

Reviewer #1: Yes

Reviewer #2: Yes

5. Review Comments to the Author

• Reviewer #1: Review PONE-D-20-27373

This manuscript presents results of an interesting study on habitat selection by an endemic subspecies of long-tailed vole. The authors used radio telemetry to determine used locations, which were compared to available locations. At the 3rd order scale they compared GIS variable, while at the 4th order scale they compared field collected vegetation data. Research on habitat selection of poorly understood species such as this are much needed. However, I found a number of problems with the presentation of the information that made it very difficult to understand the methods and results. For instance, the authors ostensibly investigated habitat selection at the “within home range” and “patch” scale (3rd and 4th order). However, they consistently referred to the 3rd order as home range selection, which is actually 2nd order selection per Johnson (1980). The methods do not say how the available locations were selected for this scale. The microhabitat data should probably also be considered 3rd order selection (usage of various habitat components within the home range. Fourth order would normally be for specific items such as den sites or specific food items. So, I think both scales analyzed are actually 3rd order.

Another problem is that they concluded that their results reflected fundamentally different habitat selection for long-tailed voles in the Pinaleno Mountains, in comparison with other populations where long-tailed voles live in presence of other competitively dominant vole species. There is no evidence for this conclusion. Importantly, they conflate their results at the 3rd or 4th order of selection with generalized statements about habitat at the 1st or 2nd order. You cannot extrapolate from one scale to another. Second, the habitat conditions described by their results are consistent with other places that long-tailed voles are known to occur, including places with other vole species. To make such a comparison they would need to have similar habitat selection results for another population of voles where other species also exist.

The discussion was very difficult to follow as it seemed to jump all over the place often not identifying what scale was being discussed. I had a very difficult time following the arguments in that section due to the writing problems. I suggest a more streamlined discussion that more concisely describes 3rd order habitat selection in this population voles (without conclusions about completion) and describes the differences in habitat selection between the sexes. Other points don’t seem very well justified to me.

• We have made a number of changes in our discussion to hopefully address the concern of this reviewer and have added sections heading to provide further structure. 

38-42 There is a bit of redundancy in these sentences. 

Thank you for bringing attention to this, lines 39 through 42 were re-worded and combined to reduce redundancy. 

76 If you are talking about the geographic range, then you are discussing a feature of the species, not the individual. A species should be referred to with the Latin name or “the long-tailed vole”, which denotes the species. If you are talking about multiple individuals that actually exist out in nature then you can refer to long-tailed voles. It is not correct to use long-tailed voles to refer to the species. 

We appreciate this clarification, we have edited ‘long-tailed voles’ to read ‘The long-tailed vole’.

82 Missing a period. Also, is there a reason to say monocots and dicots? What other herbaceous plants are there? 

Thank you for pointing these out, we added a period and replaced ‘monocots and dicots’ with herbaceous plants.

92 Populations in New Mexico are just as far south. 

The mammalian species account for Microtus longicaudus states that this species occurs at elevations of 8000 ft and above (Smollen and Keller 1987). The southernmost mountain range in New Mexico that contains Microtus longicaudus are the Sacramento Mountains (Findley and Jones 1962) in New Mexico. The southernmost point that is above 8000 ft for each mountain are as follows:

o Pinaleño mountains AZ latitude: 32.6136

o Sacramento mountains NM latitude: 32.6343

The difference in latitude shows the New Mexico population is over 6 km north latitudinally than the Arizona population.

96 In the Southwest they seem to be found in relatively moist locations and are mainly riparian (e.g., Findley et al 1975), although they also occur in openings within mixed coniferous forest that allows some herbaceous cover (e.g., Lehmkuhl et al 2008 Northwest Science 82, Wampler et al. 2008 SWNAT 53, Sullivan and Sullivan 2018 Crop Protection 112). 

Thank you for your comment on this. We restructured this sentence to reflect the variety of habitats in which long tailed voles are found. Also, ‘white-bellied vole’ was changed to ‘long-tailed vole’.

97 Something wrong with the wording 

Thank you for bringing this to our attention, the sentence was re-worded.

100 missing hyphen in white-bellied. 

Thank you for pointing out this typographical error we have added the hyphen in white-bellied.

107-109 But you don’t have a comparison with a situation where they occur with a congeneric competitor. You already made it clear that this population lacks another species of vole. I think you should delete “in the absence of congeneric competition”. I also don’t think your study design allows you to address this point. The most you can say is that your results reflect habitat selection at the third order in your study area. You don’t know how that would differ in presence of another species.

Thank you for this comment, we have removed “in the absence of congeneric competition” and reworded some of the sentence to better describe the habitat we intended.

108-109 Can you state what you predict the difference will be?

We appreciate bringing this point of clarification up. We have added some specific predictions to line 109.

114 According to the USGS place names database, Mount Graham is a specific summit of the Pinaleno Mountains. I suggest deleting Mt. Graham here unless one of your study sites is on Mount Graham, or work the name into the study area description another way. 

Thank you for your concern on this, the name ‘Mt. Graham’ was deleted to avoid confusion. 

129 What is meant by “negative road impacts”?

Examples of negative road impacts were included to improve clarity.

139 What do you mean by “depending on the saturation of the soil? 

Certain sections of the stream were diffused and bog-like. In these areas the traps were placed further away from the main channel of the stream to ensure the traps were on dry ground. This information has been added to line 141 for reader clarification. 

150 how did you get an accurate total length measurement on a live animal? Why were these measurements taken? What about mass? 

We added mass to this sentence and removed data that was not presented in the manuscript.

157 A location once per day per animal? 

Animals were located at least 5 days per week. Every animal was located once per day that we were in the field, with one animal receiving several points throughout the day spaced 1 hour apart. These additional details have been included in line 157 for reader clarification.

159 The “collar was slipped” is wildlife slang; please rephrase. Can you rephrase to avoid word predated? How did you know they had been killed by a predator? Did you find them dead and partially consumed? The “Signal was lost” is slang. 

Thank you for catching this, we have removed all jargon from this section and re-worded where needed. We feel that keeping the phrase “the vole was predated” is a more concise message than rephrasing to something of the effect “the vole was killed by a predator”.

159-160 It is still not clear to me if you recorded more than one location per animal each day. 

Every animal was located once per day that we were in the field (Monday-Friday), with one animal receiving several points throughout the day spaced 1 hour apart. These additional details have been included in line 157 for reader clarification.

157-161 How did you determine the location of the animal? Triangulation? Homing? Did you evaluate telemetry system error?

Animal locations were determined via homing as stated in line 158. We then recorded the animal location on a handheld GPS unit. We added clarification about GPS point averaging to address the issue of location error in line 163.

167 I don’t know how you located the vole. Did you see it? IF not, now can you be sure its location? 

Prior to collar deployment we tested our ability to locate collars in various situations by placing collars in known areas (bare ground, in grass, in holes). We did this to familiarize ourselves with the equipment and ensure confidence in our locations. Visuals on voles were common but did not always happen. If a visual was not obtained this was noted on our data sheet. If the animal was tracked to the same location 3 times in a row with no visuals it was assumed that the collar had fallen off and a thorough search was conducted to recover the collar, if possible, and those 3 previous points were removed from the dataset.

174 and 179 Is canopy cover the same as overstory cover? 

As far as we are able to tell these terms appear to be synonymous. Some authors use overstory to describe parts of the canopy (Wojtkowski 2008) and vis versa (Gold 2020). Some authors also use the combined term “canopy overstory” (Dellasala 2019).

1. Paul A. Wojtkowski, 4 - Agrobiodiversity, Editor(s): Paul A. Wojtkowski, Agroecological Economics, Academic Press, 2008, Pages 45-72.

2. Michael A. Gold, Anthromes—Temperate and Tropical Agroforestry, Editor(s): Michael I. Goldstein, Dominick A. DellaSala, Encyclopedia of the World's Biomes, Elsevier, 2020, Pages 107-116.

3. Dominick A. DellaSala, “Real” vs. “Fake” Forests: Why Tree Plantations Are Not Forests, Editor(s): Michael I. Goldstein, Dominick A. DellaSala, Encyclopedia of the World's Biomes, Elsevier, 2020, Pages 47-55.

174 I do not understand how the vegetation sampling was collected. Were all these variables collected at the location or on transects or a plot? 

All variables were collected at all known vole locations and random points as stated in line 176 ” At all known and random locations, we recorded understory cover, canopy cover, vegetation composition (categories: bare ground, coarse woody debris, grass, forb, fern, log, sedge, stump, rock, rush, shrub, tree, water)”. 

Understory cover-lines 183-188: “We used a 2.5 cm x 100 cm cover pole marked in 2 cm increments to measure understory cover at the center of each location [49]. We recorded the height of any obscuring vegetation from the four cardinal directions at a distance of 4 m and a height of 1 m, with any vegetation taller than 80 cm classified as 100% understory cover. We calculated percent understory cover by taking the average from all four measurements at each location, divided by the total height of the cover pole”

Canopy cover-lines 189-190: “To calculate canopy cover we followed the standard equation for a convex spherical densiometer and applied the correction factor for the 17-dot variation for each point”

Vegetation composition-lines 191-194: “We used a 1 m2 quadrat centered at each location to characterize vegetation composition and percent cover through visual inspection [51]. For any woody species within the 1 m2 quadrat, we used diameter at breast height (DBH) to categorize live and dead woody species as either shrubs (woody plants < 10 cm DBH) or trees (≥ 10 cm DBH [52])”

178 At what distance did you consider obscuring vegetation? It is not clear how you measured this.

We recorded the height of any obscuring vegetation from the four cardinal directions at a distance of 4 m and a height of 1 m. This clarification has been added into lines 180 and 181.

180 Here you mention averaging 4 measurements but you did not explain how you took them. 

Thank you for bringing this to our attention, the explanation of how the 4 measurements were obtained have been edited for clarity on lines 180 and 181. The new sentence reads “We recorded the height of any obscuring vegetation from the four cardinal directions at a distance of 4 m and a height of 1 m, with any vegetation taller than 80 cm classified as 100% understory cover”.

181 Did you take the densitometer reading in the 4 cardinal directions? 

Yes, we took densitometer readings in the four cardinal directions. The standard equation for a convex spherical densiometer using the 17-dot variation requires readings from the four cardinal directions (Strickler 1959). 

183-184 centered at the location? How did you measure composition and percent cover? 

Yes, we took these measurements centered on the location. These were assessed through visual inspection using a quadrat. These clarifications have been added, thank you for bringing this to our attention.

184 What is the plot? Do you mean the 1 m square plot?

Thank you for bringing this to our attention. Yes, we meant the 1 m square quadrat. We have edited the sentence to read ‘within the one m2 quadrat” to improve clarity. 

190 Why did you include animals with so few locations? 

Thank you for your concern on this. We evaluated three groups: All voles, males, and females. The number of locations an individual had was not a limiting factor for our models because all points were pooled into a larger dataset per group. Pooling data across individuals and then including a random effect for individual ID is ideal in low sample size situations while still controlling for individual variation (Gillies et al. 2006). This information has been added to like 200-202. At the 3rd order, we needed to constrain the available locations within an individual’s home range. The reason we included animals with at least 5 locations in these models, was because that was the minimum number of locations required by the adehabitatHR package in R.

194 I think you mean within home range scale, at least that is what I think you said you were going to evaluate in the introduction. 

Thank you for identifying this issue. All references to ‘home range scale’ have been changed to ‘within home range scale’ to assist with clarity.

194 where were the random points drawn from? 

All the random locations were generated with ArcMap tools within an individual’s MCP home range. We have added this clarification to the sentence.

196 What date did NDVI come from and why? 

The imagery the NDVI was generated from was from late summer 2017. This imagery was used because it was the most recent and highest resolution imagery available. Also, with our study being conducted during the same time of year we felt that this imagery was as representative as could be. This information has been added to line 208 and 209.

197 What were the sorts of water sources in the remote sensed imagery. Often voles are associated with moist to wet spots that would not show up on imagery or maps. You trapped along creeks. Were they perennial? 

Both creeks that we trapped near were ephemeral and visible on a USFS waterways layer for the Coronado National Forest. Additionally, a small seep that was located nearby was marked via GPS and digitized in the distance to water portion of our analysis. Despite the two creeks being designated as ephemeral, throughout the course of our study, they retained water. This information and citation have been added to lines 208 and 209.

205-206 I did not see mention of quadrates in the methods. Are these cover classes for percent cover? 

Thank you for your concern. This sentence was removed however, this issue has been corrected in the methods.

211-220 To me it seems that both of your scales are really third order, selection of sites within the home range. Your patch scale seems like description of microhabitat. Your other scale seems like within home range (not home range), although I cannot determine where you drawing the available points from. If within home range, I think your available points should be drawn from the MCP of each individual.

Johnson 1980 describes 3rd order as the usage of various habitat components within the home range and 4th order as the actual procurement of food times if the 3rd order determines a feeding site. Johnson goes on to say that these orders can be divided more finely. We argue that the 4th order does not exclusively pertain to food items and can contain selection for micro habitat as other researchers have done (e.g. Compton et al. 2002, Sprague and Bateman 2018). At the 3rd order our available points were randomly generated withing each individuals MCP home range; this information was added to line 203. At the 4th order the corresponding available point was located 9.8 m away in a randomly generated direction from the known point as stated in lines 176-180. 

214 within home range scale? 

All references to ‘home range scale’ have been changed to ‘within home range scale’ to match terminology defined by Johnson 1980.

218-219 Where are the models? How many? How rationalized? The sentence implies the same models for each scale. Do you not expect different selection at different scales

Thank you for bringing this to our attention. We have included the reasoning we based our a priori models on with this addition on line 223: ‘based on previous research of habitat selection of long-tailed voles’. Additionally, we have changed the wording to be clear that we tested different models for each scale and how many models we tested. 

224-225 I assume the male and female numbers are adults. How did you assign age class? Did you collar juveniles? Did you record sex on juveniles? 

Adults were determined by weight (anything above 27 g). Juveniles were not collared or sexed. On lines 230 and 231 we state, “we collared and tracked 31 adult voles” but we have added clarification as well. 

228-229 clarify what you mean by unknown but confirmed mortality. I think you mean you know it was dead but don’t know why. Collar slipping is slang. I think you mean that the collar fell off the animal. What does missing mean? Do you mean that you were unable to find a signal? 

Thank you for bringing this to our attention. This section has been changed to read ‘unknown cause but confirmed mortality’ and ‘missing’ has been changed to ‘no signal’ to assist with clarity for the reader.

230 What do you mean by loss of vole?

Thank you for your concern on this section. We have added the explanation “(i.e. no longer able to have data collected)” for clarity. 

222 You used individual as random effect, but I don’t know the sample size of male and females included in the analyses. 

Thank you for bringing this to our attention. There were 9 males and 19 females used in this analysis. This information has been added to lines 196 and 197.

235 by group do you mean gender? 

Thank you for your concern on this, the groups we tested our models with were all voles, males, and females. This has been clarified for the readers on lines 228 and 229.

237-238 Avoid using “shows” and “showing”. 

Thank you for your input on this, these terms have been edited throughout the manuscript.

249 I don’t think you mentioned standardizing these variables in the methods, although perhaps I missed it.

Thank you for pointing out this deficiency, we have added the sentence ‘We standardized all covariates prior to running models’ to line 216 and 217.

256 Ideally I’d like to see the PCA results perhaps in supplemental material. 

We decided to remove all mention of the PCA from the manuscript since it was unused and only caused confusion. Our intention in running a PCA was to cut down our 12 covariates into a smaller number of principal components. However, the results of our PCA showed we needed 9 principal components to account for 87% of the variation in the data. Originally, we included PC1 in our models because it accounted for 17% of the variation in data. We replaced any models that included grass and bare ground with PC1 as these were the covariates that PC1 represented. However, none of our top models for any group included PC1 based on delta AICc. In an effort to cut down on confusion we decided to remove all mention of our PCA analysis from this manuscript. 

255 So did you not use the PCA? I am confused by what is being presented in this section. 

Please see above response for concerns regarding PCA analyses.

258 This sentence does not read correctly—too many “models”. 

Thank you for bringing this to our attention, we reworded the sentence.

260-263 Why not say that the Beta coefficients indicated positive selection for … and avoidance for …. 

Thank you for this suggestion, we changed the wording of this sentence to reflect this comment.

Supplement Table 1. In looking at this table I am confused about your choice of variables and a priori hypotheses. Is the global model the top model? Are these all the models? I don’t think I understand why the PCI is used versus the other variables in the models? For instance, was log and grass also cover classes? I also see no rationale for these models.

These are all the models we ran at the 4th order. We did not run a global model in our analysis, we ran a priori models based on important factors identified in previous studies of long-tailed voles. This new addition was addressed in a previous comment on line 223. Additionally, all reference to PC1 have been removed to avoid confusion.

287 what do you mean by they showed varying degrees of selection?

Thank you for bringing this to our attention, we re-worded this line to be clearer.

290-291 Long-tailed voles are commonly captured in wet meadows and other herbaceous systems even when other voles are present. I’m not sure you can make this conclusion. Your analysis was at the home range (or perhaps within home range) and patch scales, while the habitat data you are contrasting with pertains to landscape or macro scale. I do not think it is surprising that long-tailed voles select the herbaceous environments in their surroundings. The same is true elsewhere in their range. In reflecting on this and looking back at your study area and trapping transect locations I really cannot get a feel for the environment in which the study was done. Was a mosaic of forest and open areas?

We have amended our statement to read “Our data contrast with some previous studies” and added the additional reference of Anich and Hadly 2013 to further support our statement. Randall 1978, Randall and Johnson 1979, and Anich and Hadly 20013 found that long-tailed voles compete with other vole species and experience competitive exclusion or inverse space occupancy. They found that when in the presence of other vole species long-tailed voles are rarely found in tall grass meadows and are relegated to the ecotonal tree-line areas. 

 Our study area is a mosaic of mostly closed canopy forest and open grassy meadows as well as areas of patchy tree canopy cover with an understory consisting of bare ground, forbs, ferns, and grasses. 

297-298 There are a number of studies, mainly in the Midwest, that discuss the importance of roadside environments for voles and other species that select dense herbaceous vegetation. The idea is that these areas collect more moisture resulting in more lush vegetation.

This is interesting and we have incorporated a reference in the paragraph on line 341.

286-308 long paragraph with lots of different topics

Thank you for bringing this to our attention. We have broken this paragraph up.

312-313 Selection for downed logs and coarse woody debris paints a different set of conditions than grassy areas.

Thank you for this observation, this is something we wanted the readers to know. Because there may be confusion, we have re-worded the sentence to make this clearer. To clarify, voles selected for cover types of downed logs and coarse woody debris over other cover types at the patch scale.

315 long-tailed voles occur in relatively open areas. When they occur in forested ecosystems, their local distribution is in openings in the forest where there is higher herbaceous ground cover. Later, you also say that they select high herbaceous cover. You can only get high herbaceous cover if the sun is able to hit the ground, which does not happen in dense forests.

We apologize for our use of the term “open areas” by this we meant areas of bare ground and sparse herbaceous growth. We have corrected this in lines 330. Additionally, we ensured this issue was addressed throughout the manuscript.

319-320 I don’t think you can say your results support the idea that they experience competitive exclusion from high quality vole habitat. You did not test that.

We agree, this statement and the sentence was removed.

321-322. What is open habitat dominated by woody vegetation? I don’t understand what you mean by that. 

Thank you for identifying this issue, we removed the word ‘open’.

323-324 This is a perfect description of the habitat of the long-tailed vole, but it holds true for almost any other mountain range where other species of voles also occur. I don’t think your description of habitat has to do with presence of the other species of vole. Otherwise, why would two species of vole so frequently co-occur together to the point that you catch them both on the same trap line?

You surveyed for voles along creeks in a generally forested area in the mountains. This is an ideal situation for long-tailed voles. It provides moist areas with herbaceous vegetation in openings and meadows within the forest. You can find long-tailed voles in any mountain range in this same conditions, regardless if there are other species.

Randall 1978, Randall and Johnson 1979, and Anich and Hadly 20013 found that long-tailed voles compete with other vole species and experience competitive exclusion or inverse space occupancy. They found when in the presence of other vole species long-tailed voles are rarely found in tall grass meadows and rarely caught in the same trap lines as other vole species. At the patch scale we found that white bellied voles were selecting for areas with high amounts of grassy cover. We have edited these lines to highlight the differences this population exhibits.

1. Randall JA. Behavioral mechanisms of habitat segregation between sympatric species of Microtus: Habitat preference and interspecific dominance. Behav Ecol Sociobiol. 1978;3(2):187–202.

2. Jan R, Johnson RE. Population densities and habitat occupancy by Microtus Longicaudus and M. Montanus. J Mammal. 1979;60(1):217–9.

3. Anich PS, Hadly EA. Asymmetrical competition between Microtus montanus and Microtus longicaudus in the Greater Yellowstone Ecosystem. Am Midl Nat. 2013;170:274–86.

338-339 It could also mean they select the location of home ranges based on different characteristics.

Thank you for this comment, we agree it could mean that. However, since we did not evaluate selection at the second order (home range scale), we would not feel comfortable incorporating this statement into this manuscript. 

349 And higher water requirements during lactation. 

In this sentence we believe the phrase “and increased energetic needs” includes these requirements and in an effort to keep this sentence as succinct as possible we did not incorporate this edit. Additionally, we discussed higher water requirements during lactation in line 365.

352-353 Again, I think you are confusing scales as I don’t see your results different from other locations where the species occurs.

Thank you for bringing this to our attention. To clarify our statement, we included “within home range” to line 368.

354 There is a danger of extrapolating third and fourth order selection to higher order selection such as geographic distribution. You don’t know what first and second order selection are.

We appreciate this insight; we have chosen to remove the mention of predictive use maps.

358 Are they imperiled? SGCN directs funding as is not. 

In Arizona this subspecies is a Species of Greatest Conservation Concern as stated in line 100 and 101 (Arizona’s state wildlife action plan: 2012–2022). We have restated their Arizona status here for clarification.

Reviewer #2: This is a very interesting paper that examines space use in a less-studied species of vole, The All Voles model showed that voles avoided areas of bare ground and and areas far from roads and water, Females followed the same pattern with the addition of areas with high understory cover. Females tend to remain in their terriroty near thier nest sites. Males avoided bare ground but chose areas furthre fron road s and water. The authros try to fir their data to the promiscusous/polygynous mating systems, I found this fit most tenuos as the authros really have no direct data of interatiions between and within the sexes and their statments about mating systmes comes mainly from a revew chapter by Jerry Wolff (1985).

Thank you for your concern on this topic. There is very little information on this species in regard to intraspecific social interactions. Other studies have shown there can be pronounced sex differences in feeding behavior and habitat use in other species (Clutton-Brock and Iason 1987; Newsome 1980). We feel the data we have, shows a notable difference between male and female resource selection. This combined with information from Wolff 1985 we felt comfortable suggesting white-bellied voles have either promiscuous or polygynous mating systems. We exclude the possibility of a monogamous mating system because we would expect little to no differences in resource selection in a monogamous mating system . We have modified the wording to make it clear that we believe that our results ‘suggest’ that monogamy is less likely than the polygamous systems.

1. Clutton-Brock TH, Iason GR, Guinness FE. Sexual segregation and density-related changes in habitat use in male and female Red deer ( Cervus elaphus ). J Zool. 1987;211:275–89.

2. Newsome AE. Differences in the diets of male and female red kangaroos in central Australia. Afr J Ecol. 1980;18(1):27–31.

6. PLOS authors have the option to publish the peer review history of their article (what does this mean?). If published, this will include your full peer review and any attached files.

Do you want your identity to be public for this peer review? For information about this choice, including consent withdrawal, please see our Privacy Policy.

Reviewer #1: No

Reviewer #2: No

---

## [Editor Report · Decision Letter 1]

27 Oct 2020

Resource selection of a montane endemic: sex-specific differences in white-bellied voles (Microtus longicaudus leucophaeus)

PONE-D-20-27373R1

Dear Dr. Dutt,

We’re pleased to inform you that your manuscript has been judged scientifically suitable for publication and will be formally accepted for publication once it meets all outstanding technical requirements.

Kind regards,

Bi-Song Yue, Ph.D

Academic Editor

PLOS ONE

---

## [Editor Report · Acceptance letter]

30 Oct 2020

PONE-D-20-27373R1 

Resource selection of a montane endemic: sex-specific differences in white-bellied voles (*Microtus longicaudus leucophaeus*) 

Dear Dr. Dutt:

I'm pleased to inform you that your manuscript has been deemed suitable for publication in PLOS ONE. Congratulations! Your manuscript is now with our production department. 

Kind regards, 

on behalf of

Dr. Bi-Song Yue 

Academic Editor

PLOS ONE